# Pulse Oximetry Imaging System Using Spatially Uniform Dual Wavelength Illumination

**DOI:** 10.3390/s23073723

**Published:** 2023-04-04

**Authors:** Riaz Muhammad, Kay Thwe Htun, Ezekiel Edward Nettey-Oppong, Ahmed Ali, Dae Keun Jeon, Hyun-Woo Jeong, Kyung Min Byun, Seung Ho Choi

**Affiliations:** 1Department of Biomedical Engineering, Yonsei University, Wonju 26493, Republic of Korea; riaz@yonsei.ac.kr (R.M.); kaythwetun90@gmail.com (K.T.H.); ezekieledward@yonsei.ac.kr (E.E.N.-O.); alee@yonsei.ac.kr (A.A.); 2Department of Electrical Engineering, Sukkur IBA University, Sukkur 65200, Pakistan; 3Mediana, R&D Center, Wonju 26365, Republic of Korea; dkjeon@mediana.co.kr; 4Department of Biomedical Engineering, Eulji University, Seongnam 13135, Republic of Korea; hwjeong@eulji.ac.kr; 5Department of Biomedical Engineering, Kyung Hee University, Yongin 17104, Republic of Korea; 6Department of Electronics and Information Convergence Engineering, Kyung Hee University, Yongin 17104, Republic of Korea; 7Department of Integrative Medicine, Major in Digital Healthcare, Yonsei University College of Medicine, Seoul 06229, Republic of Korea

**Keywords:** pulse oximetry imaging system, spatially uniform dual wavelength illumination, hyperspectral imaging system

## Abstract

Pulse oximetry is a non-invasive method for measuring blood oxygen saturation. However, its detection scheme heavily relies on single-point measurements. If the oxygen saturation is measured at a single location, the measurements are influenced by the profile of illumination, spatial variations in blood flow, and skin pigment. To overcome these issues, imaging systems that measure the distribution of oxygen saturation have been demonstrated. However, previous imaging systems have relied on red and near-infrared illuminations with different profiles, resulting in inconsistent ratios between transmitted red and near-infrared light over space. Such inconsistent ratios can introduce fundamental errors when calculating the spatial distribution of oxygen saturation. In this study, we developed a novel illumination system specifically designed for a pulse oximetry imaging system. For the illumination system, we customized the integrating sphere by coating a mixture of barium sulfate and white paint inside it and by coupling eight red and eight near-infrared LEDs. The illumination system created identical patterns of red and near-infrared illuminations that were spatially uniform. This allowed the ratio between transmitted red and near-infrared light to be consistent over space, enabling the calculation of the spatial distribution of oxygen saturation. We believe our developed pulse oximetry imaging system can be used to obtain spatial information on blood oxygen saturation that provides insight into the oxygenation of the blood contained within the peripheral region of the tissue.

## 1. Introduction

Oxygen saturation, also referred to as blood oxygen saturation (SpO_2_), is a crucial physiological parameter, along with heart rate, breathing rate, blood pressure, and body temperature [1,2]. The measurement of blood oxygen saturation provides the relative amount of oxygen present in the blood and serves as an indicator of whether a person has an adequate supply of oxygen [3]. A decrease in oxygen levels can cause long-term damage to individual cells, the brain, and the heart, as well as short-term malfunctioning of other important human organs [4,5]. Therefore, accurate and frequent measurements of oxygen saturation are essential for monitoring the well-being of vital organs and for the early detection of potential health issues. For measuring blood oxygen saturation, a variety of options including invasive and non-invasive methods are available from both the literature and commercial devices. In an invasive SpO_2_ estimation method, the levels of oxyhemoglobin (HbO_2_) and deoxyhemoglobin (Hb) can be determined through a blood gas analysis where blood samples are collected, and the number of oxygen-binding hemoglobin is counted. Although this technique is accurate, it is costly, time-consuming, and invasive.

Pulse oximetry is commonly used as a non-invasive method for measuring the blood oxygen saturation (SpO_2_) and the heart rate of patients in hospitals and homes [6]. Pulse oximetry measures the percentage of oxygenated hemoglobin in the arteries [7]. The measurements determine the supply of oxygen to peripheral tissues and the efficiency of oxygenation in the pulmonary alveoli [8,9]. During detection, the interaction of light with the area of measurement is achieved via transmission or reflection methods. In the transmission method, light incidents on one side of the targeted peripheral region, and the transmitted light is detected by a photodetector placed either on the opposite side or side-by-side with the light source, whereas the light source and photodetector are located on the same side of the targeted region in the reflection method [10,11]. Photoplethysmogram (PPG) signals, generated by shining red and near-infrared light on the skin, provide information on both oxygenated and deoxygenated hemoglobin and are commonly used to quantify blood oxygen saturation. This is achieved by evaluating the ratio of absorption coefficients of oxyhemoglobin to deoxyhemoglobin, which reflects the saturation level of oxygen in the blood [12].

Conventional pulse oximetry methods are limited to single-point measurements, and hence the measurements are influenced by the shape of illumination, spatial variations in blood flow [13], and skin pigment [14,15]. To overcome these challenges, imaging systems that measure the distribution of blood oxygen saturation have been demonstrated. Humphreys and Markham [16] presented a contactless system equipped with a CMOS camera to capture photoplethysmography (PPG) signals at 760 nm and 880 nm, respectively. Two lights were alternately illuminated on the fingertip to detect rhythmic arterial pulsation. Subsequently, the results were compared to those of a conventional contact-based pulse oximeter. Although the results were comparable, the SNR of the PPG signal was highly dependent on the location in the peripheral region of the tissue. Al-Naji et. al. [17] have used the green and red channels of a digital camera to estimate SpO_2_ values. However, the estimated SpO_2_ values are greatly influenced by the lighting conditions.

Moreover, previous camera-based pulse oximetry imaging systems relied on two different patterns of red and near-infrared illuminations [18,19,20]. When spatial profiles of the input red and near-infrared illumination are different, the ratio between transmitted red and near-infrared light also differs across space [20]. The mismatch between the red and near-infrared illumination profiles can introduce fundamental errors [21]. Therefore, spatially uniform dual wavelength illumination is required to eliminate the occurrence of artifacts due to spatially varying light sources.

In this study, we developed a novel illumination system specifically designed for use in a pulse oximetry imaging system. The illumination system developed in this study created two identical patterns of red (630 nm) and near-infrared (940 nm) illuminations that are spatially uniform. To achieve this unique illumination, we customized an integrating sphere by coating its interior with a mixture of barium sulfate and white paint.

An integrating sphere, also referred to as the Ulbricht sphere, is an optical device used for the spatial integration of radiant flux, and thus has a diffusing effect or the uniform scattering of light. It is composed of a hollow sphere with small openings which serve as input and output ports for light. The interior of the sphere is coated with a highly reflective material to obtain diffusely reflecting walls, which enables multiple scattering reflections of light within the sphere. Incident light rays are scattered multiple times within the sphere, effectively averaging out any directional variations in the light output. Consequently, a more homogenous illumination is achieved. The operation of the integrating sphere is described by a mathematical expression of radiance. The general expression for the radiance, *L*, corresponding to an illuminated diffuse surface with an input flux, *Φ_i_*, is given as:(1)$$L=\frac{\Phi_{i}p}{\pi A}$$
where *p*, *π*, and *A* are the reflectance, the total projected solid angle from the diffuse surface, and the illuminated area, respectively [22]. Considering monochromatic and unidirectional irradiation, the reflectance is described by the ratio of the reflected flux, independent of direction, to the incident flux from a differential solid angle about the *θ* and *ϕ* direction, where *θ* is the polar angle of incidence measured from the normal of the surface, and *ϕ* is the azimuthal angle of incidence measured in the plane of the surface from an arbitrary reference [23]. Reflectance is expressed in terms of the reflection-distribution function *f (θ*, *ϕ′*; *θ′*, *ϕ′*) integrated over half-space. The function evaluates the intensity of the radiance corresponding to the reflected flux in the direction *θ′*, *ϕ′* per unit radiant flux incident in the differential solid angle about the *θ* and *ϕ* direction. A surface is considered diffuse for a continuous *f* and perfectly diffuse for a constant *f* [23].
(2)$$p(\theta ,\;\phi )=\int_{0}^{2\pi }\int_{0}^{\pi /2}f(\theta ,\phi \prime ;\;\theta \prime ,\;\phi \prime )\sin\theta \prime \cos\theta \prime d\theta \prime d\phi \prime$$

For an internally illuminated integrating sphere, both the numerous surface reflections and losses via the port apertures required for the input flux are considered in the radiance equation [22].
(3)$$L=\frac{\Phi_{i}}{\pi A_{s}}\;\times \;\frac{p}{1-p(1-f)}$$
where *f* is the port fraction. The radiance equation has two components; the first component is identical to the general expression for radiance (Equation (1)), and the second component is termed the sphere multiplier [22]. The sphere multiplier, *M*, is a unitless quantity that accounts for the radiance increase resulting from multiple reflections.
(4)$$M=\frac{p}{1-p(1-f)}$$

The value of *M* is determined by both the reflectance of the sphere surface and the port fraction. The port fraction is expressed as a function of the input port area, *A_i_*, the output port area, *A_o_*, and the sphere wall area, *A_s_*.
(5)$$f=\frac{A_{i\;}+A_{o}}{A_{s}}$$

The developed pulse oximetry imaging system incorporates this illumination system to capture high-precision spatial and temporal data. Using a hyperspectral imaging approach, we obtained spectral information from different locations within the peripheral region based on the acquired spatio-temporal data. Hyperspectral imaging is a multivariate imaging technique. An image made of *I* rows and *J* columns acquired over *K* variables is a classical multivariate image. The commonly utilized variables are wavelengths, but other variables can be used as well [24]. By interpolating between the coinciding points of the dual images captured due to the dual wavelength light source, a spectrum is generated to characterize different locations within the peripheral region. Utilizing the hyperspectral imaging approach and fundamental theory of pulse oximetry, a spatial map of the blood oxygen saturation was produced. Moreover, the developed system overcomes the limitations of existing systems, where the spatial intensity distribution of red and near-infrared lights is not uniform, yielding approximate blood oxygen saturation measurements.

## 2. Materials and Methods

### 2.1. System Design

A transmission pulse oximetry imaging system was developed for measuring the spatio-temporal distribution of blood oxygen saturation over time. As indicated in Figure 1a, the developed technology has two major components: an imaging system, and a light source. The imaging system consisted of a CMOS camera, a zoom lens, an integrating sphere, and light emitting diodes (LEDs) as the light source. Red (630 nm) and near-infrared (940 nm) LEDs were used as input light sources for the integrating sphere. In addition, an Arduino microcontroller was used to generate trigger signals for the measurements. As demonstrated by the signal patterns in Figure 1a, the LEDs alternate in response to the trigger signals with a corresponding exposure time for the camera. The system repeats this operation to capture spatio-temporal images for data processing. A hyperspectral imaging approach was used to obtain spectral information from the captured images at wavelengths of 630 nm and 940 nm. Thus, the spatio-temporal distribution data of blood oxygen saturation is acquired. Figure 1b shows the front and top view of the experimental set-up of the pulse oximetry imaging system. The following sections further elaborate on the materials and components utilized in developing the imaging system.

#### 2.1.1. Imaging Setup

The camera, responsible for capturing light from the peripheral region through reflection or transmission, plays a critical role in developing a pulse oximetry imaging system. The performance of the camera immensely affects the quality of the acquired images and, ultimately, the accuracy of the blood oxygen saturation measurement [25]. A monochrome CMOS camera (model acA1300-60gmNIR, Basler AG, Ahrensburg, Germany) featuring a 1/1.8 in CMOS sensor with a maximum resolution of 1280 × 1024 square pixels of length 5.3 μm and 12-bit encoding was used. The camera was connected to a PC via IEEE 1394 and utilized a manually adjusted C-mount zoom lens (model 87-536 0.15X-0.5X Non-Telecentric Lens, Edmund Optics, Barrington, NJ, USA) with a focal length range of 50–179.8 mm and an aperture of *f*/2.8–*f*/22, to focus on the tissue under investigation. Upon receiving trigger signals, the camera captures a set of frames at 28 frames per second using a global shutter technique.

#### 2.1.2. Illumination System

An ideal pulse oximetry imaging system requires a light source that exhibits low spatial intensity variations and maintains a consistent illumination pattern over time and different wavelengths. In traditional pulse oximetry imaging, inhomogeneous illumination arises from the inconsistent illumination profile of the input light across various peripheral locations. This results in imprecise blood oxygen saturation measurements [26]. To overcome these issues, we created an illumination system by customizing an integrating sphere to produce uniform illumination at red and near-infrared wavelengths.

The illumination system design aimed to balance light distribution and brightness while conforming to established guidelines for the proper operation of integrating spheres. In accordance with these guidelines, commercially available integrating spheres typically have diameters ranging from 50 mm to 250 mm, with openings that do not exceed 5% of the sphere’s total area and an angle of light incidence not greater than 10 degrees [27]. To maximize the light intensity at the exit port of the sphere, following the design guidelines cited before, a 50 mm diameter sphere was designed featuring 16 input ports of 4 mm diameter on its sides for LED lights and a 12 mm diameter output port on its top for uniform illumination emission. The total port area was designed to occupy 4.61% of the sphere’s surface with a light incidence angle of 10 degrees. The design of the integrating sphere was realized using a computer-aided design (CAD) and fabricated from white Polylactic Acid (PLA) material using a 3D printer (model Flashforge Guider II, Zhejiang Flashforge 3D Technology Co., Ltd., Jinhua, China) with a filament thickness of 1.75 mm.

#### 2.1.3. Light Emitting Diodes (LEDs)

The transmission-type pulse oximetry imaging system requires a light source having both visible and infrared lights with stable and uniform illumination. Red and near-infrared lights have been used for pulse oximetry systems, since they maximize the penetration for optical imaging techniques [28]. The illumination source used in this study consisted of eight red and eight near-infrared light-emitting diodes (LEDs) inserted into an integrating sphere from all four sides (refer to Figure 1a). The chosen number of LEDs per wavelength ensured stable, uniform, and intense illumination at red (630 nm) and near-infrared (940 nm) wavelengths.

#### 2.1.4. Reflectance Coating Material

Barium sulfate (BaSO_4_) is a white, crystalline powder known for its high reflectance and has been used as a reference standard in optical measurements [29]. It offers a cost-effective alternative to more expensive white standards and provides uniform reflectance across the visible and near-infrared wavelength range [28]. Additionally, the crystalline structure of BaSO_4_ results in high scattering properties [30]. Hence, it is a suitable choice for the in-house integrating sphere.

A mixture of Barium sulfate (97.5–100%, SAMCHUN Chemical, Seoul, Republic of Korea) and white paint (model PXI453901/4L, NOROO Paint and Coatings, Anyang, Republic of Korea) at a 50:50 volume ratio with 20% water content was prepared. The mixture was stirred and allowed to settle for 2 h to achieve a uniform mix. The interior of the integrating sphere was then coated with three layers, each with a specified drying time of 10 h, to ensure a consistent thickness. The cross-sectional view of the coated integrating sphere is shown in Figure 1c—upper-left. To evaluate the uniformity of light scattering on the sphere’s inner surface, green and red laser pointers with wavelengths of 532 nm (22 mW output power) and 650 nm (8 mW output power), respectively, were utilized. The lasers were focused on a sectioned piece of the sphere, and the distribution of scattered light was recorded (see Figure 1c—bottom-left and right). The evaluation of scattered light on the inner surface of the cut piece of the integrating sphere revealed a uniform distribution of light, which was indicative of uniform reflection scattering across the entire surface due to multiple scattering events (see Section 3.1 for a complete analysis of the illumination pattern).

The reflectance spectra of the mixture of BaSO_4_ and white paint were measured and compared with Spectralon white standard (model USRS-99-010, Labsphere Inc., North Sutton, NH, USA) using an integrating sphere (model 2P4/M, Thorlabs, Newton, NJ, USA) and a compact CCD spectrometer (model CCS200/M, Thorlabs, USA). The results showed that a mixture of BaSO_4_ and white paint provides a reflection of 87% (see Figure 1d), which is acceptable for multiple scattering purposes. The high reflectance value achieved by the mixture of barium sulfate and white paint confirms its suitability for use as the reflective material in the integrating sphere for efficient reflection and scattering of light inside the integrating sphere.

#### 2.1.5. Microcontroller for Generating Trigger Signals for Camera and Light Source

An Arduino microcontroller was used to control both the red (630 nm) and near-infrared (940 nm) LEDs and the camera, as shown in Figure 1a. A separate power supply was required for the 630 nm LEDs due to the higher current draw, which is shown in Figure 1b. This power supply provided a regulated 12 volts and was switched on and off by a trigger signal generated by the Arduino through a transistor (model MPS 2222A, ON Semiconductor, Scottsdale, AZ, USA). In contrast, the 940 nm LED array was powered directly from the trigger signal generated by the Arduino, eliminating the need for an additional power supply. Using the Arduino microcontroller to generate trigger signals ensures precise control of both red (630 nm) and near-infrared (940 nm) LEDs for accurate imaging.

### 2.2. Fundamental Theory of Dual Wavelength Pulse Oximetry

Blood oxygen saturation (SpO_2_) is the percentage of oxygen in the arterial blood, and it is determined using the following formula [31]:(6)$${\mathrm{SpO}}_{2}=\frac{\left[{\mathrm{HbO}}_{2}\right]}{\left[{\mathrm{HbO}}_{2}\right]+[\mathrm{Hb}]}\;\times 100\%$$
where [HbO_2_] and [Hb] are the concentrations of hemoglobin with oxygen and without oxygen, respectively.

The specific absorption coefficients of oxygenated (HbO_2_) and deoxygenated hemoglobin (Hb) have distinct spectral dependence [32], as included in Figure 1e. The difference between these coefficients allows for the quantification of each constituent, forming the basis of pulse oximetry [33,34]. At wavelengths below 800 nm, the Hb-specific absorption coefficients are higher than the HbO_2_ one, while in the region above 800 nm, the HbO_2_ absorptions dominate. As shown in Figure 1e, red light at 630 nm is absorbed by deoxygenated hemoglobin (Hb) more than oxygenated hemoglobin (HbO_2_), whereas near-infrared light at 940 nm is absorbed by HbO_2_ more than Hb. Upon transmission or reflection through tissue, the red or near-infrared light is attenuated due to absorption by biological molecules and scattering by larger tissue structures. The modified Beer-Lambert Law [35] is used to express the intensity of the transmitted or reflected light from the skin tissue to quantify the light absorption by hemoglobin as follows:(7)$$I_{t\;}=I_{o}e^{-CD\alpha }$$
where *I_t_* is the intensity of the transmitted light, *I_o_* is the intensity of incident light, *C* is the concentration of the sample, α is the absorption coefficient of the tissue, and *D* is the optical path length (i.e., the distance traveled through the sample). The calculation of SpO_2_ based on pulse oximetry assumes that the pulsatile component of optical absorption is due to the pulsating flow of arterial blood, while the non-pulsatile component stems from non-pulsating arterial blood, venous blood, and other tissues. The ratio of absorbances is calculated using the red (630 nm) and near-infrared (940 nm) time signals in the following equation:(8)$$R_{630}=\frac{{\mathrm{AC}}_{630}}{{\mathrm{DC}}_{630}}\;R_{940}=\frac{{\mathrm{AC}}_{940}}{{\mathrm{DC}}_{940}}\;\mathrm{RR}=\frac{R_{630}}{R_{940}}$$
where AC is the pulsatile component, DC is the non-pulsatile component, and RR is the ratio of ratios of the absorbances at the two wavelengths. Previous studies [36,37] have shown that the pulsatile and non-pulsatile components correspond to the standard deviation and mean color intensities of the red and near-infrared frames, respectively. A nearly linear relationship exists between SpO_2_ values and RR [38]. Therefore, the value of SpO_2_ is estimated from RR according to the equation:(9)$${\mathrm{SpO}}_{2}=(m\;\times \;\mathrm{RR})+c$$
where *m* and *c* are empirically determined through linear regression for each volunteer, *m* is the slope of the estimated regression line, and *c* is the y-intercept [39]. The empirical approximation technique is used to correct errors in the measured values caused by the assumption of only two substances in the light path [40].

### 2.3. Image Acquisition

The monochrome CMOS camera was controlled by an Arduino and took images with a resolution of 1280 × 1024 pixels at a frame rate of 28 frames per second. Images were captured every 36 milliseconds when either LED was on. This led to a camera trigger rate of 28 times per second, resulting in a data acquisition rate of 14 frames per second for each wavelength. Camera exposure time was set at 18 ms to optimize the contrast-to-noise (CNR) ratio [41,42]. The recorded frames were sent to a personal computer (PC) via IEEE 1394 data communication and stored automatically in external memory as a sequence of time-series images in TIFF format for future image processing. A buffer was employed to store the images and prevent data loss during transfer to the PC. All experiments were conducted in a completely dark environment to minimize noise.

### 2.4. Image Processing

In this study, image processing was used as a method to calculate the spatial distribution of blood oxygen saturation through a pulse oximetry imaging system that employs a dual wavelength uniform illumination light source. We have used MATLAB (Version R2022b, MathWorks, Natick, MA, USA) to process the time series images.

Once a set of recordings was successfully acquired, the raw image frames were divided into discrete regions of interest (ROIs) to produce a new set of reduced frames, where the value of each pixel in the new frame was set as the average of all the pixel values within each ROI. Though compromising the spatial resolution, such a procedure significantly improved the signal-to-noise ratio. Averaging the pixel values within each ROI is a better sampling technique that preserves more spatial information as compared to capturing reduced images of the peripheral region. Herein, the ROI size was set at 20 × 20 pixels. This resulted in a reduced frame size of 64 × 52 pixels, yielding time series PPG signals at each pixel position across a sequence of frames.

The mean of the pixel values (16-bit, ranging from 0 to 65,536) within an ROI region was computed for each frame using the following equation:(10)$$S(t)=\frac{\sum_{i-1}^{i-X}\sum_{j-1}^{j-Y}s(i,j,t)}{X\cdot Y}$$
where X and Y represent the width and height of the ROI region, respectively, and t is the frame sequence. These averaged values were used to obtain the imaging-PPG signal at the corresponding wavelength (see Figure 2).

The PPG signals were used as the physiological indicators to analyze the pulsatile changes in blood volume and blood oxygen saturation levels at various locations over different wavelengths. Each of the two PPG signals was divided into a 10-s subset, and the AC and DC components of the PPG signals were obtained using average peak-to-peak and mean values, respectively, from each of these subsets (see Figure 2). The minimum points at each signal were interpolated across all spatial locations to remove the baseline from the PPG signals, and the DC component was subtracted from the AC component. Consequently, the RR values were calculated from Equation (8) using the determined AC and DC components of the PPG signals. Finally, the blood oxygen saturation (SpO_2_) values were extracted from the calculated RR values at each spatial location using Equation (9).

## 3. Results and Discussion

The pulse oximetry imaging system was assessed to determine its efficacy to measure the spatio-temporal distribution of blood oxygen saturation over time. The performance of the developed system was evaluated in three phases. First, characterization of the illumination pattern of the designed integrating sphere to ensure uniform illumination. Next, analysis of the captured images to ensure photoplethysmogram (PPG) waveforms are consistent for all the locations within the peripheral region. Finally, acquisition of the spatio-temporal information of the distribution of blood oxygen saturation to determine physiological parameters including heart rate, respiratory rate, and oxygen saturation of tissues. The outcomes were compared to existing pulse oximetry imaging systems in literature and commercial devices.

### 3.1. Illumination Pattern Assessment

Herein, a key focus was on developing an illumination system with a highly uniform profile that remains constant over time to achieve precise measurements of blood oxygen saturation. By utilizing the working principle of an integrating sphere, a virtual light source was developed to overcome this challenge. The integrating sphere uniformly distributes light over the entire inner surface of the sphere, creating a highly homogenous spatial illumination pattern. The dimensions of the designed integrating sphere are illustrated in Figure 3a. Substituting the dimensions and the determined reflectance (87%) of the integrating sphere into Equations (4) and (5), the port fraction, *f*, and the sphere multiplier, *M*, were calculated to be 0.04 and 5.279, respectively.

The determined sphere multiplier value indicates that the fabricated integrating sphere ensured multiple reflections of the incident lights, resulting in an overall increase in radiance. Thus, the fabricated integrating sphere used in the developed pulse oximetry imaging system was efficient in yielding high radiance. The uniform distribution of light within the sphere ensures a consistent uniform profile independent of time and changes in wavelength. As shown in Figure 3b, the output light from the designed integrating sphere was uniformly distributed over the entire outport port area for both input light sources at 630 nm and 940 nm. The realized uniform light profile is also illustrated in Figure 3c. The 3D profiles of the output light demonstrate the homogenous spatial illumination pattern obtained from the designed integrating sphere.

In addition, the acquired homogeneity ensures that the light source is consistent with the input trigger signals. This consistency is also critical for accurate measurements of SpO_2_ in pulse oximetry, as any discrepancies between the light source and the input trigger signals can lead to inaccuracies in the measurements. Figure 3d presents an assessment by utilizing a photodetector and a camera to demonstrate the compatibility between the light source and the input trigger signals. The timing diagram from the photodetector confirmed that fluctuations in the timing or intensity of the light source relative to the input trigger signals were trivial. Thus, the light source was highly consistent with the input trigger signals, with minimal variations. The images captured by the camera further demonstrated consistency by revealing that the homogenous spatial illumination pattern produced by the integrating sphere was capable of uniformly illuminating the sample.

Additionally, we characterized the transmission pattern of our illumination system. To achieve this, we placed a thin diffuser sheet over the output port of the integrating sphere. Next, we alternately turned the input LEDs ON and OFF and captured images of the transmitted light through the diffuser at two wavelengths—630 nm and 940 nm (see Figure 4a). Subsequently, we calculated the intensity difference by subtracting the 940 nm profile from the 630 nm profile and plotted it over space and time (see Figure 4b). We also calculated the mean variance of the difference over time (see Figure 4c). The integrating sphere ensured a uniform illumination profile over the dual wavelengths in both space and time, resulting in improved image quality. Thus, this uniform and consistent illumination system enables the light to penetrate through the skin and provide the spatial information of the light absorbed by the blood, resulting in spatial distribution measurements of blood oxygen saturation.

### 3.2. Photoplethysmography

An experimental test was conducted by imaging a fingertip (refer to Appendix A). Figure 5a illustrates the relative position of the imaged illuminated area of the fingertip. The mean values of the pixels contained in the highlighted regions of the fingertip are plotted against time. The waveforms in the projected PPG graph are inverted, and the received pixel light intensity is replaced with light absorption on the vertical axis. Light absorption is directly proportional to the peripheral arterial pressure waveform, whereas the received light intensity is inversely proportional. In the analyzed PPG signal, the systolic peaks and diastolic troughs are easily distinguishable. Moreover, the dicrotic notch—an inflection in the waveform caused by the abrupt closure of the aortic valve—is distinctly visible. The outlined observations are consistent with expected physiological behavior and further support the validity of the PPG signal as a measure of cardiovascular activity.

The signal-to-noise ratio (SNR) plot suggests that the PPG signals obtained from different spatial locations are consistent and have a relatively constant SNR (see Figure 5b). This consistency across spatial locations, distinct pulse amplitude, and phase information demonstrates the efficiency of the pulse oximetry imaging system and further confirms the uniformity of the illumination used. A high-quality PPG signal which reflects the amount of transmitted or reflected light that penetrates the skin is crucial for obtaining accurate physiological parameters such as heart rate, respiratory rate, and blood oxygen saturation (SpO_2_). Fourier spectra of the red (630 nm) and NIR (940 nm) PPG signals are depicted in Figure 5c. The pulsatile component, evident at 1.19 Hz (71.4 bpm), is profound in both spectra. The respiratory component, demonstrated at 0.25 Hz (15 BR/min), is also apparent in both signals. The acquired results are comparable to the results obtained from a Patient monitor (model M40, MEDIANA Co., Ltd., Seoul, South Korea), which recorded 1.23 Hz (73 bpm) and 0.26 Hz (16 BR/min) for the pulsatile and respiratory components, respectively.

Table 1 provides a comparative analysis between the proposed system and other pulse oximetry imaging systems. Previously demonstrated pulse oximetry imaging systems have used LED arrays [39,43] or LED rings [16,44] that relied on two different illumination profiles for dual wavelengths. When the spatial profiles of the two illuminations are different, the ratio between transmitted light will also differ. Such a mismatch between the illumination profiles can introduce fundamental errors when calculating the spatial distribution of blood oxygen saturation. In contrast, our illumination system created two identical patterns of red and near-infrared illumination, of which patterns were spatially uniform with standard error of only 1.2% and 1.3%, respectively (see Figure 3b). We utilized this illumination system to capture spatial and temporal data, from which we obtained spectral information at different locations within the peripheral region.

### 3.3. Spatio-Temporal Information of Blood Oxygen Saturation

The spatial distribution of blood flow as a function of time (i.e., the spatio-temporal data) and the derived PPG waveforms are shown in Figure 6a for red illumination and in Figure 6b for NIR illumination. The spatial and temporal characteristics of blood flow in the fingertip were analyzed by utilizing the images captured under the red (630 nm) illumination. The processed images, captured over the period of time during the experiment, demonstrated a clear connection between the pulsatile blood flow in the fingertip and the cardiac cycle (see Figure 6a). To further analyze the relationship between the cardiac cycle and pulsatile blood flow in the fingertip, the captured time-series images of the fingertip during near-infrared (940 nm) illumination were processed after removing the DC component. Near-infrared imaging offers increased tissue penetration depth due to reduced photon absorbance and scattering. As a result, the use of near-infrared illumination is advantageous in pulse oximetry imaging systems [45]. The resulting images, shown in Figure 6b, highlight the spatial variations in the pulsatile blood flow.

#### SpO_2_ Quantification

To evaluate the performance of our illumination system, we calculated the spatial distribution of blood oxygen saturation (SpO_2_) in the fingertip using the fundamental theory of pulse oximetry. In pulse oximetry, the AC component corresponds to the maximum peak of the PPG signal. The time-varying spatial maps of pulsatile blood flow obtained using both red (630 nm) and near-infrared (940 nm) images correspond to the amplitude variations of a PPG signal (see Figure 6a,b). The processed images under red illumination show the maximum peak (AC_630_) of the PPG signal, which corresponds to high blood volume at t = 684 ms. Similarly, under near-infrared illumination, the AC_940_ corresponds to high blood volume at t = 720 ms. We normalized the AC components with DC components at these temporal points. Last, we calculated SpO_2_ distribution using Equation (4), and by utilizing calibrated *m* and *c* values of −21.56 and 110.66, respectively. The average SpO_2_ level measured by our system was 97.15% (see Figure 6c). For comparison, the SpO_2_ level of the subject was also measured using the Patient monitor (model M40, MEDIANA Co., Ltd., Wonju, Republic of Korea), and a value of 99% was recorded. To minimize potential calibration errors, it is recommended that the system be tested on several volunteers. The developed pulse oximetry imaging system successfully measured the pulsatile variation in blood and SpO_2_ levels of the peripheral region. Table 2 compares the SpO_2_, heart rate, and respiratory rate data from our proposed system and a commercially available Patient monitor device (model M40, MEDIANA Co., Ltd., Wonju, Republic of Korea).

Table 3 compares the measurement systems for blood oxygen saturation based on the parameters analyzed. Single channel pulse oximeters only monitor temporal changes in transmitted light at a single point to calculate the blood oxygen saturation. If blood oxygen saturation is measured at a single location, the measurements are influenced by the shape of illumination and spatial variations in blood flow [2]. However, our proposed imaging system can capture the spatial and temporal changes in transmitted light at various spatial locations over the red (I_630 nm_ see Figure 6a) and near-infrared (I_940 nm_ see Figure 6b). Additionally, the proposed system can provide the spatial distribution and temporal change of SpO_2_, as demonstrated by Figure 6c. On the other hand, the single-channel pulse oximeter provides only the temporal changes of I_630 nm_ and I_940 nm_, and the SpO_2_ at a single location. The table highlights the superior performance of our proposed system over single-channel pulse oximeters in terms of spatial and spectral data acquisition.

## 4. Conclusions

In this study, we developed a camera-based pulse oximetry imaging system that employs a spatially uniform dual wavelength light source to observe the spatial-temporal changes in blood oxygen saturation over the peripheral regions, such as the fingertip or earlobe. We used a pulse oximetry technique in combination with a hyper-spectral imaging approach to characterize different locations within the peripheral region. Our results showed that both 630 nm and 940 nm images contain physiological information about the blood flow within the peripheral region of the tissue under investigation. We combined the processed red and near-infrared images to estimate the blood oxygen saturation at individual locations within the peripheral region. The measured physiological parameters were comparable to those of a commercial Patient monitor.

Our pulse oximetry imaging system offers several advantages over conventional single-channel pulse oximetry methods. Our imaging technique eliminates limitations related to inconsistent illumination profiles between the two wavelengths in the peripheral region and provides rich spatio-temporal information about blood oxygen saturation at different locations. The development of an integrated device utilizing this imaging technique holds significant potential for improving biomedical applications, such as wound monitoring and real-time, long-term monitoring of blood oxygen saturation, where accurate tissue oxygen content information is critical.

## Figures and Tables

**Figure 1 sensors-23-03723-f001:**
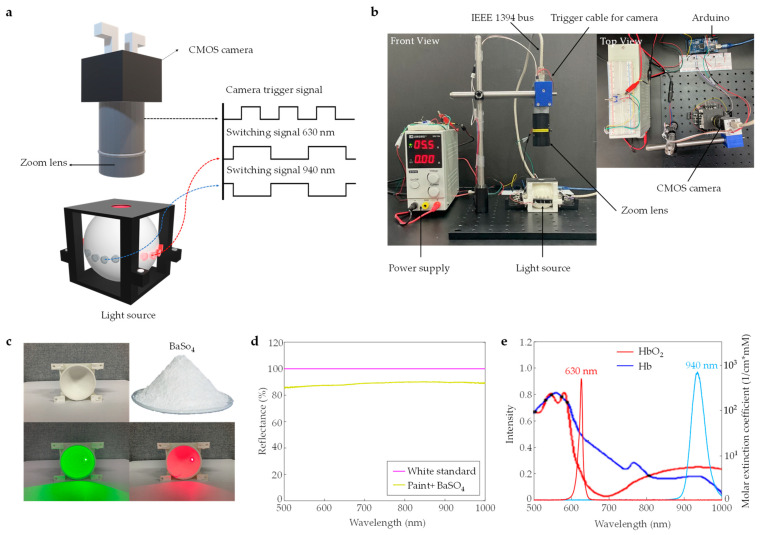
Optical/electrical configuration and materials of pulse oximetry imaging system: (**a**) Illustration of the proposed system; (**b**) Front and top views of the experimental setup with hardware trigger, PC connection, camera, light source, and power supply; (**c**) Cross-sectional view of the 3D printed integrating sphere coated with the reflectance material (upper left), BaSO_4_ powder (upper right), green laser (lower left) and red laser (lower right) projected on the integrating sphere; (**d**) Reflectance spectra of white standard and the coating material (paint + BaSO_4_) used for the integrating sphere; (**e**) Spectra of 630 nm and 940 nm wavelength LEDs, along with the deoxyhemoglobin and oxyhemoglobin absorption spectra within the 500 nm to 1000 nm wavelength range.

**Figure 2 sensors-23-03723-f002:**
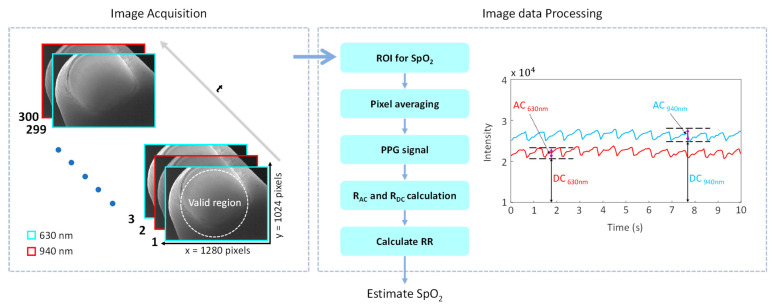
Image acquisition and data processing for SpO_2_ calculation: Image acquisition. The image data is captured alternately at 630 nm and 940 nm in a time series of frames, with a resolution of 1280 × 1024 pixels for each frame. The region above the output port of the integrating sphere was considered a valid measurement area. Image data processing. The panel illustrates the steps for calculating the SpO_2_ values and displays the AC and DC components of the 630 nm and 940 nm photoplethysmogram (PPG) signals.

**Figure 3 sensors-23-03723-f003:**
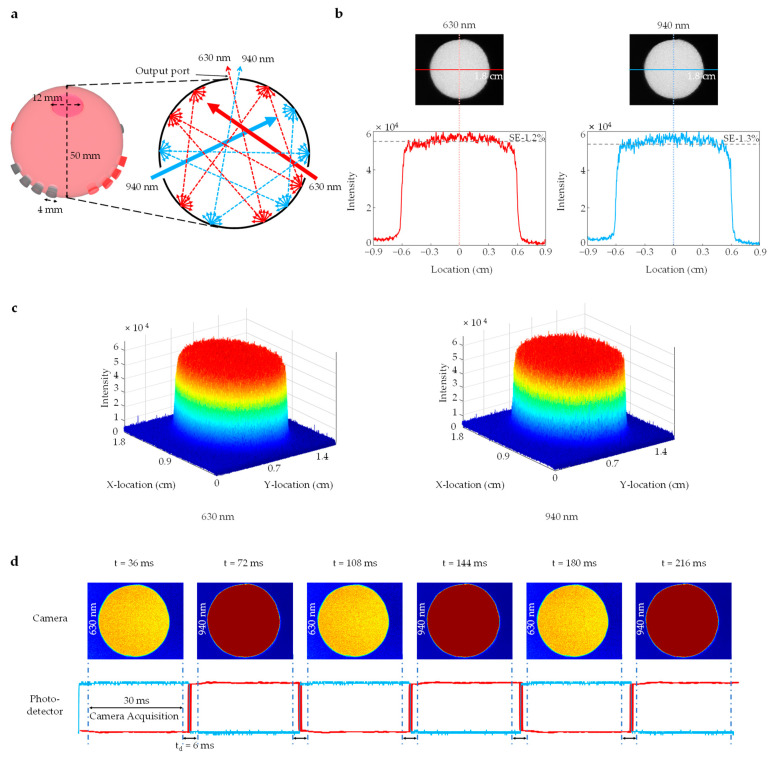
Characterization of illumination profile: (**a**) The physical dimensions of the integrating sphere (the integrating sphere has a diameter of 50 mm with 4 mm holes for 16 LEDs and a 12 mm light output port) and demonstration of light-scattering patterns in the integrating sphere; (**b**) The figure displays the illumination intensity of 630 nm (left) and 940 nm (right) that comes out from the output port with the standard error of 1.2% for 630 nm and 1.3% for 940 nm at 1.8 cm field of view (FOV); (**c**) The 3D illumination profile of the 630 nm and 940 nm at 1.8 cm (FOV) were generated using MATLAB R2022b; (**d**) Frames of 630 nm and 940 nm wavelengths were captured alternately, producing homogenous spatial patterns over time. The LED ‘ON’ and ‘OFF’ timing is synchronized with camera acquisition, with a total delay of 6 ms (3 ms before camera exposure starts and 3 ms after the camera frame is captured).

**Figure 4 sensors-23-03723-f004:**
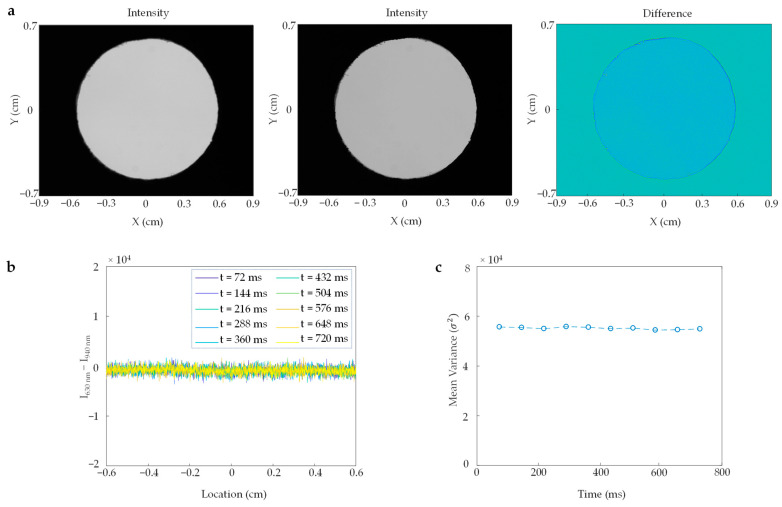
Quantification of uniformity and similarity of red and near-infrared illuminations: (**a**) Transmitted light illumination intensity at 630 nm (**left**), 940 nm (**center**), and the intensity difference between 630 nm and 940 nm (**right**) emitted from the output port of the integrating sphere. To capture the images of the transmitted light illumination intensity, a diffuser was placed over the output port of the sphere with a field of view (FOV) of 1.8 cm on the *x*-axis and 1.4 cm on the *y*-axis; (**b**) The spectrum displays the change in intensity difference between 630 nm and 940 nm at a central field of view of 1.2 cm over a period of time ranging from 72 ms to 720 ms; (**c**) The plot depicts the mean-variance in intensity difference between 630 nm and 940 nm over time.

**Figure 5 sensors-23-03723-f005:**
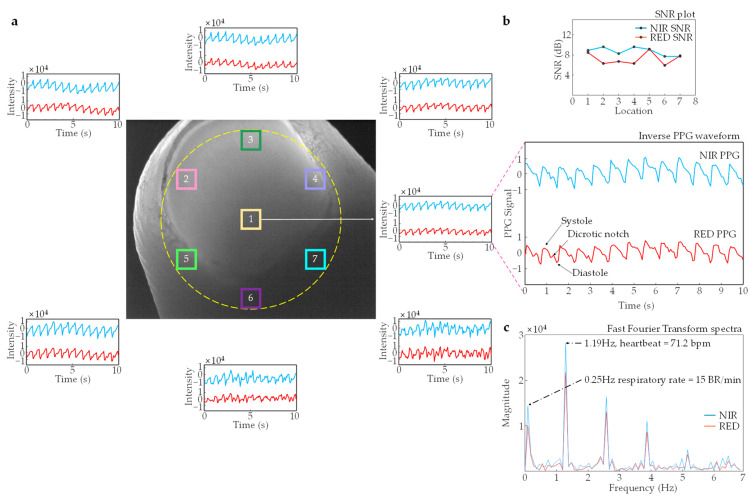
Measurement of photoplethysmogram using pulse oximetry imaging system: (**a**) The photoplethysmogram (PPG) signals acquired at various spatial locations are displayed, demonstrating time-varying spatial features at different wavelengths; (**b**) the signal-to-noise ratio (SNR) plot for both the red and near-infrared wavelengths; (**c**) Fast Fourier Transform (FFT) spectra of the RED (630 nm) and near-infrared (NIR) (940 nm) PPG signals are shown, highlighting the heart rate at 1.19 Hz (71.2 bpm) and respiratory rate at 0.25 Hz (15 BR/min).

**Figure 6 sensors-23-03723-f006:**
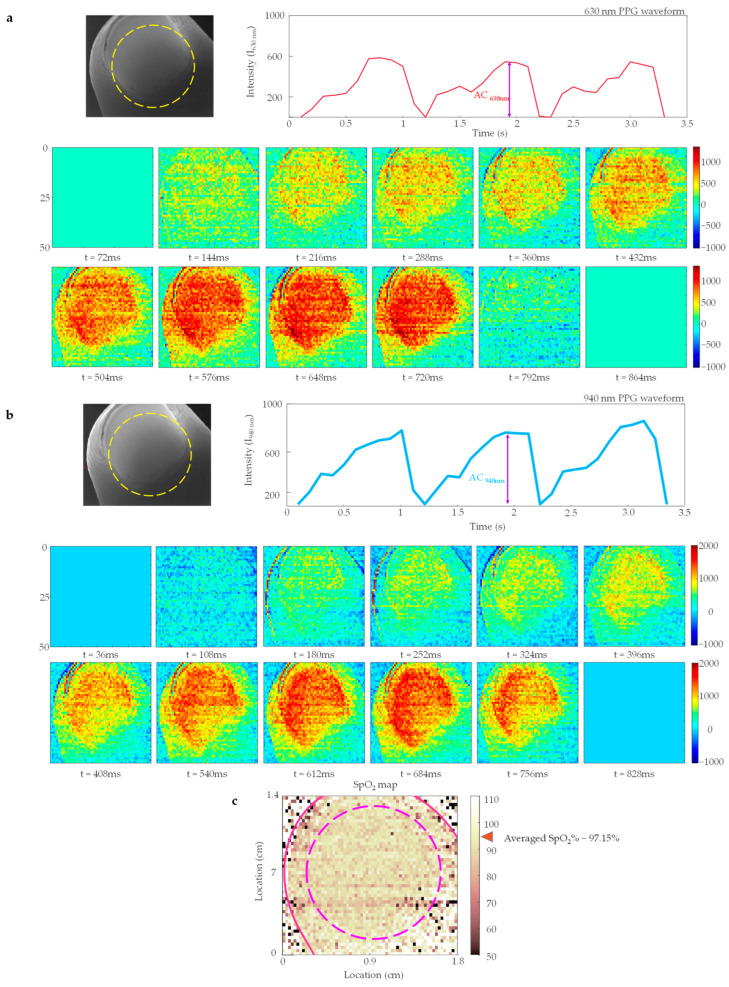
Spatio-temporal assessment of blood oxygen saturation using pulse oximetry imaging system: (**a**) The image illustrates the validated area of the finger illuminated by red (630 nm). The raw Photoplethysmogram (PPG) waveform is shown for three cardiac cycles. The figure also shows the processed 12 images for a single cardiac cycle showing spatial variation in pulsatile blood flow; (**b**) The image illustrates the validated area of the finger illuminated by red (940 nm). The raw Photoplethysmogram (PPG) waveform is shown for three cardiac cycles. The figure also shows the processed 12 images for a single cardiac cycle showing spatial variation in pulsatile blood flow (for a video demonstration of three cycles, refer to Appendix A); (**c**) The SpO_2_ map represents the spatial distribution of blood oxygen saturation within the validated region of measurement. The average SpO_2_ measured for the subject was 97.17%.

**Table 1 sensors-23-03723-t001:** Comparison of our illumination system with other reported illumination systems.

Imaging System	Type of Illumination	Signal-to-Noise Ratio
Our proposed system	Dual wavelength (630 nm & 940 nm) Integrating sphere uniform light source (SE—1.2% & 1.3%)	SNR ratio range: 6–10
Non-contact Imaging Plethysmography [44]	Dual wavelength (520 nm & 660 nm) LED ring light source	N/A
Non-contact monitoring of blood oxygen saturation using camera and dual wavelength imaging system [39]	Dual wavelength (610 nm & 880 nm) LED array light source	N/A
A CMOS camera-based system for clinical photoplethysmography applications [16]	Dual wavelength (520 nm & 660 nm) LED ring light source	N/A
Non-contact Imaging Photoplethysmography [43]	Dual wavelength (650 nm & 880 nm) LED array light source	N/A

**Table 2 sensors-23-03723-t002:** Comparison of physiological parameters from our system with a commercially available patient monitoring system.

	Our Proposed System	Commercial Patient Monitor
SpO_2_ (%)	97.15	99
Heart rate (bpm)	71.4	73
Respiratory rate (BR/min)	15	16

**Table 3 sensors-23-03723-t003:** Comparison of our system’s specifications with a single-channel pulse oximetry system.

System Parameter	Our Proposed System	Single Channel Pulse Oximetry System
Spatial resolution	106 × 106 µm	N/A
Spectral data at 630 nm, I_630 nm_	Provide spatial distribution and temporal change of I_630 nm_; Series of I_630 nm_ images visualizes pulsatile blood flow on region of interest (Figure 6a).	Provide temporal change of I_630 nm_ at a single location
Spectral data at 940 nm, I_940 nm_	Provide spatial distribution and temporal change of I_940 nm_; Series of I_940 nm_ images visualizes pulsatile blood flow on region of interest (Figure 6b).	Provide temporal change of I_940 nm_ at a single location
Blood oxygen saturation level, SpO_2_	Provide spatial distribution and temporal change of SpO_2_ (Figure 6c)	Provide temporal change of SpO_2_ at a single location

## Data Availability

The data presented in this study are available from the corresponding authors upon request.

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
