# Peer review of "Pulse Oximetry Imaging System Using Spatially Uniform Dual Wavelength Illumination"

_sensors, 2023, doi:10.3390/s23073723_

Round 1
Reviewer 1 Report
Riaz et al. presents in the manuscript titled “Pulse Oximetry Imaging System Using Spatially Uniform Dual Wavelength Illumination” a built-in house CMOS-based pulse oximeter utilizing spatially uniform dual-wavelength illumination to produce a homogenous illumination to obtain spectral information at different locations within the peripheral region. Important part is related to 3D printed, BaSO4 coated integrated sphere, which construction process was sufficiently described. Previously named sphere, together with CMOS camera and Arduino controlled LEDs are used to snap hypespectral-like images of human fingernail, which are calculated into PPG signals.
Authors suggest that the system could be used for monitoring tissue oxygenation in patients with diseases associated with poor blood circulation. The idea is interesting but novelty of the study is not clearly explained.
A more detailed explanation how this approach achieves better accuracy, compared to a single location measurement, related to the factors such as insufficient blood flow, exposure to light, variations in skin pigmentation, and an irregular pulse should be added considering the size of the region of the interest. This explanation can signify more the advantages of this work over the conventional single channel pulse oximetry methods. In the abstract, the research question needs to be clarified. Also some key papers are missing in literature. In particular, comparison and validation of the results with recent papers would increase the overall quality of the paper.
- The introduction does not sufficiently reflect the motivation for the study, I would like more information about the currently available methods of oximetry and their shortcomings.
- The article does not contain references to important formulas, such as the general formula for determining the degree of blood oxygenation (1) and the Lambert-Beer law (2).
- The manuscript should be better finished before submission. For instance, there are many typos and different fonts (ArialMT, SimSun, Calibri, SegoUI, CambriaMath) in the manuscript that they must change based on journal template (PalatinoLinotype). Font of legend of all figures should be change to PalatinoLinotype.
- Please, be more careful with the abbreviations. For example, SpO2 was mentioned in lines 32, 38, 74, and near-infrared (NIR) in lines 27, 61, 64, and all the abbreviations need to be checked carefully.
Detailed comments:
22/23 – pulse oximetry by itself doesn’t have anything to do with artificial intelligence, better explanation needed
47 – coma not needed
51 – “requires skins to be punctured”, maybe just invasive
53 – commonly used as
57 – demonstrate pulse oximetry? what does it mean
80/81 – detector does not have to be on the opposite side, it can be also placed side-by-side
91 – not measurements but results were comparable
97 – developed technology
99 – red light not light red
113 – “system has system”, maybe imaging part? Its correct but hard to follow sentence
114 – integrating sphere is not light source
117 – synchronize is the word, not simultaneous
121 – hyperspectral imaging approach was already mentioned before but its explained just now
136 – word is missing, accuracy of estimation? calculation? measurement?
136/137 – camera description is not connected with anything, maybe authors wanted to say that camera was used
141 - What is the f? f/2.8 - f/22?
142 – focus on the tissue
142 – receipt? received triggering signal or detected triggering signal/detected rising/falling edge, etc
142 – camera capture set of frames
146 – I would not say that those wavelengths were used “historically”, its enough to say why. History have nothing to do in here, just pragmatism
Fig. 1. Description is very confusing, too much information at a time, some parts could be removed, other simplified. Moreover, there is an error in figure 1.e! Incorrectly shown spectrum of oxy and deoxyhemoglobin. The isosbestic point is located in the figure at a wavelength of 750 nm, while in the description of the figure in the article it is correctly described that the isosbestic point is in the region of 800 nm, 805 is most often mentioned in the literature. However, if we consider the spectrum of the 500-1000 nm range, then we are confused oxyhemoglobin peaks at 540 and 570 nm, as well as a deoxyhemoglobin peak at 570 nm.
199 – BaSO4 could be compared with commercially available integrating spheres. There are commercially available spheres with nearly 98% reflectance in this region. Nearly constant illumination is not the right description in here.
198 – our design or built in house
228 – Is “conjunction” correct term, transistor might be driven by micro controller?
231/232 – ambiguous sentence, could be skipped
253 - absorption co-efficient => absorption coefficient
254 - final bracket was not put on line
262 – shown, not reported
272 – it was already mentioned multiple times, Arduino is in control, it is redundant to mention it
284 – employed?
286 – used
290 – remove second “reduced frame”, redundant
ROI- why not to capture already reduced image, less data to process and sampling rate could be higher? Needs at least explanation
330 – 362 – I don’t find a reason to include all those information into results section, maybe introduction
Figure 3 A – colors of rays are changing from red to blue, thanks to figure 3, some parts of figure 1 may be removed
3D – black and white might not be best selection of colors, it gives impression that one is always on and other off
394 - figure 3. C. What software is used to generate the 3D illumination profile? Please explain, referring to the reference.
414 – custom integrating sphere-based light source? bad naming
418 – references are needed, this sentence needs prove! Compare it to something
Figure 5: Plots needs titles, then in the text thre is no need to refer to location of the plot (bottom left, example line 452).
451 – 463 – if breath rate is mentioned it should be measured by some additional apparatus to prove the point, otherwise 0.25 Hz breath rate and 1.19 Hz pulsation is assumed
464 – 477 – Please explain why nail was chosen for pulse oximetry, it is constantly named as “fingertip”, where in fact it is nail
Figure 6: once again titles
Conclusions are too broad, first paragraph could be removed.
Reviewer 2 Report
A dual Wavelength-based Pulse Oximetry Imaging System is proposed. Below a summary of comments that some of them (2, 3, 4) need to be addressed.
1-Though it is a prototype, however it requires a lot of resources for little advantages. For instance, the prototype requires 16 LEDs, a additional/separate sources was required for the red-630nm LED.
2-The software used to process data must be mentioned.
3-Authors need to compare their work with the state of the art in the same area. For instance with single and dual wavelengths illumination systems.
4-It is claimed that the developed system has advantages over the existing ones for monitoring physiological parameters for patients with diseases associated with poor blood circulation, however the provided results are for persons without any diseases.
5-It is mentioned several time about the superiority of the proposed system, however no evidence is provided to justify the above claim. Authors must clearly show show the advantages of the proposed system over the existing and/or commercially available systems.
Round 2
Reviewer 1 Report
The manuscript has been revised ass suggested, and I don’t have further requests for revision.
Reviewer 2 Report
The manuscript as a whole has been improved
The questions have been answered adequately
The claims have been supported by results